# Urbanization and Management of the Catchment Retention in the Aspect of Operation of Storm Overflow: A Probabilistic Approach

**Bartosz Szeląg [1],\*, Agnieszka Cienciała [2], Szymon Sobura [2], Jan Studziński [3] and Juan T. García [4]**

[1]    Department of Geotechnics and Water Engineering, Faculty of Environmental, Geomatic and Energy
      Engineering, Kielce University of Technology, 25-314 Kielce, Poland
[2]    Department of Geomatics, Cadastre and Real Estate, Faculty of Environmental, Geomatic and Energy
      Engineering, Kielce University of Technology, 25-314 Kielce, Poland
[3]    Systems Research Institute of Polish Academy of Sciences, 01-447 Warsaw, Poland
[4]    Department of Mining and Civil Engineering, Universidad Politécnica de Cartagena, 30-202 Cartagena, Spain
\*    Correspondence: bszelag@tu.kielce.pl

**Abstract:** This paper presents the concepts of a probabilistic model for storm overflow discharges, in which arbitrary dynamics of the catchment urbanization were included in the assumed period covered by calculations. This model is composed of three components. The first constitutes the classification model for the forecast of storm overflow discharges, in which its operation was related to rainfall characteristics, catchment retention, as well as the degree of imperviousness. The second component is a synthetic precipitation generator, serving for the simulation of long-term observation series. The third component of the model includes the functions of dynamic changes in the methods of the catchment development. It allows for the simulation of changes in the extent of imperviousness of the catchment in the long-term perspective. This is an important advantage of the model, because it gives the possibility of forecasting (dynamic control) of catchment retention, accounting for the quantitative criteria and their potential changes in the long-term perspective in relation to the number of storm overflows. Analyses carried out in the research revealed that the empirical coefficients included in the logit model have a physical interpretation, which makes it possible to apply the obtained model to other catchments. The paper also shows the use of the prepared probabilistic model for rational catchment management, with respect to the forecasted number of storm overflow discharges in the long-term and short-term perspective. The model given in the work can be also applied to the design and monitoring of catchment retention in such a way that in the progressive climatic changes and urbanization of the catchment, the number of storm overflow discharges remains within the established range.

**Keywords:** logistic regression; probabilistic model; stormwater; urbanization

---

## 1. Introduction

Urbanization of urban catchment areas and ongoing climate change are leading to an increase in the number of heavy rainfall events. This results in an increase in the instantaneous maximum flows and an increase in the volume of rainwater flowing out of catchment areas, which leads to hydraulic overloads due to overfilling of the retention reservoirs, an increase in the number of storm overflows, and urban floods due to stormwater flowing to the surface of the area. As the permissible number of discharges for the existing stormwater systems is usually exceeded [1,2], the phenomena of discharging stormwater are more frequent than the accepted standards of land drainage [3]. Therefore,

there is a need to take appropriate measures to improve the efficiency of land drainage systems. At the same time, the reduction of the number and volume of storm overflows is of key importance for the improvement of the operation of stormwater systems. This is confirmed by numerous works in this field [4–6].

In view of the above, efforts are being made to implement the principles of sustainable development at the stage of the development of catchment areas and stormwater networks. In this approach, it is crucial that the development of the catchment area and the facilities located in it does not occur chaotically, that the implemented solutions do not lead to deterioration of the quality of water in the receivers, and do not result in deterioration of the conditions for the operation of stormwater systems. For this purpose, intensive research has recently been undertaken to reduce the amount of wastewater discharged from stormwater systems. This was achieved by controlling the streams of stormwater discharged through stormwater channels [7] and by constructing and selecting optimal retention reservoir structures [8,9]. The problems related to the regulation of the stormwater flow (flow regulators using kinetic and potential energy) mentioned above are currently the subject of many studies, in which the assessment of the impact of actions taken in the operation of the stormwater system is based on the number and volume of storm overflows [10]. An alternative approach are systems [11,12] in which the installed equipment operates online, regulating in real-time the flow values in individual sections of the stormwater system. The results of simulation calculations performed confirm the high efficiency of the adopted solutions and ensure intelligent development of urban settlement units, taking into account ecological and economic aspects.

One of the possible solutions for the reduction of the stormwater stream is to reduce the amount of stormwater flowing out of particular partial catchment areas [13,14]. At present, this is often the subject of work in which various systems are considered to ensure that stormwater is retained within individual catchment areas and discharged into the ground in a controlled manner. One of the possible solutions is to build infiltration trenches and basins, retain stormwater in reservoirs, and use it for sanitary purposes [15]. In addition, green roofs, rain gardens, and similar local solutions can be used to reduce the stream in the catchment area [16–18]. Both the regulation of the stormwater stream in the drain system and the reduction of its volume in partial catchment areas lead to an increase in the retention of the catchment area, which has a key impact on the operation of storm overflows.

In order to assess the impact of the adopted solutions on the operation of storm overflows (annual number of discharges), hydrodynamic models of catchment areas are used [19,20]. The literature review [21,22] shows that one of the commonly used tools to simulate the operation of drainage systems in urban areas is the Stormwater Management Model (SWMM) program. Its development (and many other programs, e.g., MOUSE, PCSWMM, STORMCAD, etc.) is based on detailed data on catchment characteristics and data on continuous measurements of precipitation and flows. However, it is not always economically viable to perform a hydrodynamic model, especially as the simulation results may often be unsatisfactory. Moreover, during the operation of the basin it is often extended, which makes it necessary to recalibrate the changed model. From a practical point of view it is troublesome, because the data obtained on the basis of the model simulation results should be the basis for making decisions on the spatial development of the catchment area, and the change of the model makes this impossible.

Considering the above, probabilistic models were used to simulate the annual number of storm overflows [23–25]. Thorndahl and Willems [24], who used the First Order Reliability Model (FORM) method to forecast the occurrence of overflow discharge in the case of a single rainfall episode, presented one of the first works in this scope. However, this method is computationally complex. Therefore, further works were undertaken, which resulted in the application of a classification model of logistic regression to simulate the occurrence of overflow discharge [25]. This model allows for the analysis of the studied phenomenon in catchments with different physical and geographical characteristics, provided that the weighted average retention of the catchment is determined [25]. However, in this model, as well as in other models developed so far, the aspects related to the change of land use in

the catchment area were not taken into account, which is a significant disadvantage of these models, limiting their application.

Therefore, it is advisable to develop a probabilistic model for forecasting the annual number of discharges, which would take into account the possibility of assessing the impact of the urbanization of the catchment area on its functioning. The paper proposes such a model, consisting of three independent elements, i.e., a rainfall generator, a model to simulate storm overflow discharge in the case of a single rainfall episode, and the function of a change in time of the degree of imperviousness of the catchment area. To develop the model, the results of measurements from an exemplary city catchment in Kielce were used.

The research described in the paper was undertaken by the authors due to the fact that in the vast majority of currently operated stormwater systems, storm overflows are inadequately designed, in particular in relation to situations when, due to climate change, more frequent and more intense and abundant rainfall events occur, causing rapid accumulation of stormwater, to which these overflows are not adapted. As a result, there are more situations where, due to hydraulic overloads of existing overflows, stormwater from stormwater pipes is poured out and transport routes in cities and cellars in residential buildings are flooded. This, of course, creates uncomfortable situations for town citizens and generates demand for solving this problem by municipal authorities and water supply companies. The proposals for modeling storm overflows presented in the article, taking into account rainfall forecasts, make it possible to develop an Informatics Systems (IT) tool for redesigning storm overflows already in operation, and in particular for designing new stormwater systems appropriate for the requirements created by the gradual warming of the climate. It will also enable safe building of the municipal catchment area, preventing harmful spilling of stormwater during heavy rainfall. These properties of the developed model will enable safe and sustainable urban development, which will result in better quality of life and satisfaction of city residents.

## 2. Methodology

Within the framework of the conducted research, an innovative probabilistic model was proposed to simulate the annual number of storm overflow discharges, while it is also possible to make long-term forecasts. In contrast to other works [23–25], the model provides an additional possibility of adjusting the degree of imperviousness of the basin, which gives the possibility of creating sustainable development of the basin, taking into account its spatial management plans. Consequently, it allows limitation of the impact of urbanization on the number of storm overflow discharges or elimination of this by implementing stormwater management systems [26,27]. The research assumes that for the existing condition of the examined catchment area and sewage system, the annual number of storm overflow discharges will be calculated, the permissible number of which is usually determined in accordance with the binding industry guidelines in the relevant country [28–30]. The proposed probabilistic model consists of three independent elements. The first element is the logistic regression model, which allows for identification of storm overflow discharge in the case of a single rainfall episode. Another element is the rainfall generator, which allows simulation of the stochastic nature of rainfall in annual and multiannual cycles. The last element of the model is a function which accounts for changes in the degree of imperviousness of the river basin in simulation calculations over the forecast period.

The individual calculation steps of the calculation algorithm presented in Figure 1 are as follows:

(a) Determination of the logit model for the forecast of overflow discharge, which requires the separation of rainfall events in the time series of rainfall episodes, determination of changes in the characteristics of the catchment area in the period covered by the calculations, and identification of episodes of discharges by storm overflow on the basis of the measurement data;

(b) Determination of rainfall characteristics in individual episodes and determination of the dynamics of rainfall changes by determining empirical distributions and adjusting theoretical distributions to them on the basis of calculations of the Kolmogorov-Smirnov test;

(c) *K*-sampling simulations of rainfall characteristics (based on fixed theoretical distributions) in M rainfall episodes per year using the Monte Carlo Latin-HyberCube (MC + LH) method; in the case of multi-year modelling rainfall data, the number of episodes modelled in the series of rainfall shall be equal to T times the number of predicted annual series;

(d) Determination of the number of storm overflow discharges for modelling *K*-series of rainfall using a logistic regression model;

(e) The establishment of a curve describing the probability of exceeding (CDF) a specified number of discharges in a given time period;

(f) Determination of the number of discharges per year ($Z_a$) and comparison with the limit value ($Z_m$), which allows one to decide on potential change of the degree of imperviousness of the catchment area.

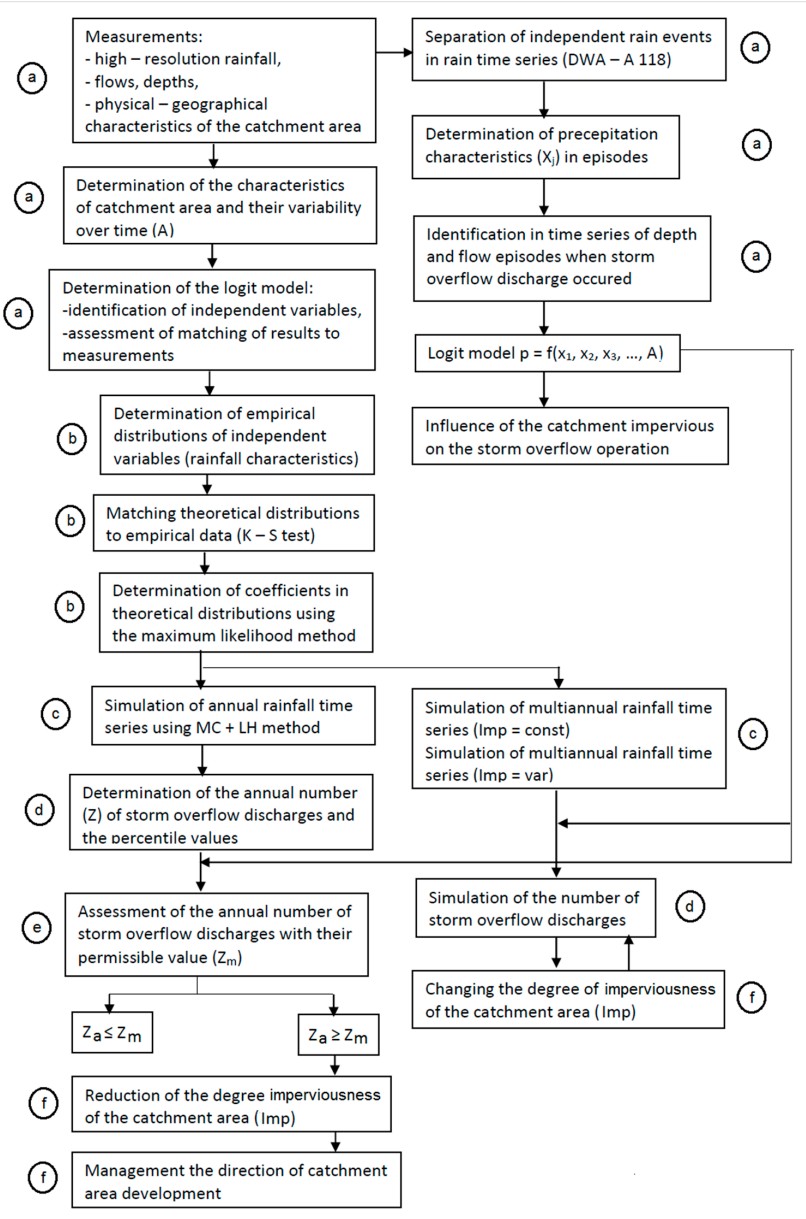

**Figure 1.** Calculation algorithm of a probabilistic model for forecasting the number of storm overflow discharges.

The paper presents the results of each calculation step of the algorithm (determination of the logit model, development of the rainfall generator, and simulation of the annual number of discharges taking into account the changing degree of the catchment imperviousness) in the example of the city catchment in Kielce.

## 2.1. Logistic Regression

The logistic regression model, also called the binomial logit model, belongs to the classification models. One of its advantages in comparison with other classification models for data analysis is the property that the result of calculations is not zero or one, but the value of probability of occurrence of a phenomenon, episode, state, or event. This fact has been widely used in many fields of science, including medicine, social sciences, and ecology [31,32]. In recent years, this model has become increasingly popular in simulating the operation of stormwater systems and the objects located in them [25,33]. However, in issues where sustainable land use problems in urban catchments are taken into account, this model has not been applied so far. The logistic regression model in general form is described by the equation:

$$p = \frac{exp\left(\alpha_0 + \sum_{k=1}^{j} \alpha_k \times x_k\right)}{1 + exp\left(\alpha_0 + \sum_{k=1}^{j} \alpha_k \times x_k\right)} \tag{1}$$

in which probability $P$ of occurrence of $s$ discharges by storm overflow for the number $N$ of rain events in a year is described by the following binomial distribution:

$$P(s) = \binom{N}{s} p_i^s \times \left(1 - p_i^s\right)^{N-s} \tag{2}$$

where: $p_i$ is the probability of occurrence of discharge in the $i$-th rainfall event; $x_1, x_2, \ldots, x_i$ are independent variables describing rainfall characteristics; $A_1, A_2, \ldots, A_m$ are independent variables describing catchment characteristics (where $m$ is the value of the degree of imperviousness assigned for following years); and $\alpha_k$ are the empirical coefficients estimated using the maximum likelihood method.

In the calculations performed it was assumed that the storm overflow discharge during a single episode takes place when the specified value of $p$ for the adopted independent variables ($x_j$, $A_i$) will not be lower than 0.50, which corresponds to the following condition:

$$\alpha_0 + \sum_{k=1}^{j} \alpha_k \times x_k \geq 0 \tag{3}$$

In this work, the simulation of storm overflow discharge was based on rainfall characteristics concerning the total amount of rainfall ($P_{tot}$) and its duration ($t_r$). Such an assumption is accepted in the works of numerous researchers [2,23–25] who have dealt with this issue. Among the independent variables describing the characteristics of the catchment area, the share of impervious surfaces (%Imp) in the catchment area was taken into account in the calculations. In relation to the works of other authors dealing with the overflow discharge forecast, it is a significant extension of the scope of application of the model obtained. This approach makes it possible to assess changes in the imperviousness of the basin over time (during the year and also in the longer term) and to plan the management of the basin (its retention) in such a way as to reduce the impact of urbanization on the quality of water in the receiver.

The following indicators were used to assess the predictive capabilities of the logit model: sensitivity (SENS) expresses the correctness of data classification in a data set covering the period when an overflow discharge occurred; specificity (SPEC) expresses the correctness of data classification in a data set covering cases when no overflow discharge occurred; and calculation error ($R_z^2$) expresses the

correctness of identification of events (a bank transfer discharge occurred or did not occur), discussed in detail in the paper by McFadden [34].

To build a logistic regression model, the results of measurements from the period 2009–2011 were used, in which out of 188 episodes of rainfall, overflow discharge occurred in 69 events. The calculations were also based on data from the period 2012–2017, in which out of 261 observed overflow episodes the overflow discharge occurred in 140 events.

### 2.2. Separation of Rainfall Events in Time Series

A key task in the construction of statistical models to simulate objects located in stormwater networks is to determine the values of precipitation characteristics that determine the observed phenomenon. Thus, at the stage of creating the model it is necessary to separate the so-called independent rainfall events [35]. The review of literature [36,37] shows that in most cases the basis for the identification of rain events is the antecendent period between the following rainfall duration events ($t_r$) and the total precepitation depth ($P_{tot}$). The analyses assumed a minimum antecendent period length of 4 h [37] and a minimum rainfall depth of 3.0 mm [38,39].

Based on the above assumptions, independent rainfall events were distinguished in the time series of rainfall (2008–2017) from the rainfall station located 2 km from the catchment boundary. The performed analyses showed that in particular years the number of events was in the range of $36 \div 61$, and their average number was 47. Parameterization of rainfall events using the total depth of rainfall ($P_{tot} = 3.0 \div 45.2$ mm) and its duration ($t_r = 20 \div 2366$ min) showed their high variability in time. Calculations made by Szeląg et al. [25] showed that the variability of rainfall depth ($P_{tot}$) and duration ($t_r$) in the analyzed municipal catchment can be described by means of Weibull distribution:

$$F(x_i) = 1 - exp\left\{-\left(\frac{x - \mu}{\gamma}\right)^{\beta}\right\}$$

(4)

where $x$ is the variable; $\beta$, $\gamma$, $\mu$ are empirical parameters of distributions; for the variables $P_{tot}$ and $t_r$ they are, respectively, $\mu = 3.0$ mm; $\beta = 0.714$, $\gamma = 5.336$, and $\mu = 12.0$ min; $\beta = 0.994$, $\gamma = 491.94$ min.

On the basis of theoretical distributions for $P_{tot}$ and $t_r$ variables and for the average annual number of rain episodes per year, a $K$ times simulation of $M$ rain events was performed using the Iman-Conover method.

### 2.3. Rainfall Generator

One of the key factors influencing the operation of stormwater systems and the facilities located in them is the variability of rainfall, which is of a probabilistic nature. The literature review [40–42] shows that a frequently used tool for their simulation are multidimensional distributions of probability density created on the basis of boundary distributions (determined on the basis of measurements) and appropriately selected dome functions connecting boundary distributions. Despite numerous implementations of this method, this solution is not easy and requires the use of complex numerical algorithms.

One of the alternatives to dome functions is the Iman-Conover method (IC), i.e., an additional algorithm used in the Monte Carlo method. In this approach, the assessment of variable correlation is the value of the Spearman correlation coefficient. Results of a simulation performed using the IC method are considered valid when:

(a) The mean values ($\mu_1, \mu_2, \dots, \mu_i)_s$ and standard deviations ($\sigma_1, \sigma_2, \dots, \sigma_i)_s$ for the variables ($x_i$) in $j$ samples do not differ by more than 5% from the values obtained from the theoretical distributions;

(b) Empirical distributions of modelled values of $x_i$ variables (in $j$ samples) are in accordance with theoretical distributions; in order to check this condition, it is recommended to use the Kolmogorov-Smirnov test;

(c)　　The value of the correlation coefficient ($R$) between dependent variables ($x_i$) obtained for MC simulation data does not differ by more than 5% from the R value determined for empirical data.

If one of the above conditions is not met, increase the number of MC simulations, recalculate, and check the above conditions. A detailed description of the assumptions and theoretical considerations concerning the IC method can be found in the works of Iman-Conover [43], Wu and Tsang [44], and Tarpantelli et al. [45], among others.

In the paper based on theoretical distributions (total rainfall depth, duration of rainfall), Iman-Conover's method for the studied catchment was used to simulate synthetic events in annual rainfall series.

### 2.4. Urbanization of Catchment Area in the Time Horizon

Simulation of the impact of the assumed degree of imperviousness of the catchment area (for the assumed calculation period, i.e., 1 year, 2 years, etc.) on the operation of stormwater systems with the use of probabilistic models is a frequently raised problem [46,47]. The results obtained in this way are useful, as they allow one to quantify the negative effects of urbanization and make appropriate decisions on how to limit uncontrolled development of the catchment area [14,48,49]. When in the time period ($T$) covered by the calculations, in the following years ($t = 1, 2, 3, \ldots, T$) the degree of imperviousness of the catchment area changes, then the analyses become more complicated. However, since the data in the form of series $z_1, z_2, z_3, \ldots, z_{N \cdot T}$ (where $p = 1, 2, 3, \ldots, N \cdot T$, and $z_p = [P_{tot}, t_r]$) obtained with the use of precipitation generators are random, subsequent values of $[P_{tot}, t_r]_p$ cannot be directly assigned to subsequent years.

Using this fact it was assumed in calculations that the subsequent values of $z_p = [P_{tot}, t_r]_p$ in the data series, including $N_1$ of precipitation episodes (where $z_p \leq N_1$), are assigned the degree of imperviousness Imp ($t = 1$), and for the subsequent episodes ($z_p > N_1$) where $T \cdot N > N > N_1 + p$, the degree of imperviousness Imp ($t = t$ *) is assigned.

## 3. Object Research

The object of the analyses was a city catchment area of 62 ha located in the south-eastern part of Kielce (Figure 2). The city covers an area of 109 km$^2$ located in Poland and is the capital of the Świętokrzyskie Voivodeship. The average population of the area in question is 21.4 people·ha$^{-1}$. The impervious areas of the catchment are pavements (8.4%), roads (17.70%), parking lots (11.20%), roofs (14.3%), and school playgrounds (1.3%). Road density in the basin is about 108 m/ha. The remaining part of the catchment area is occupied by pervious areas, i.e., lawns and other green areas. Figure 3 shows the change in the degree of imperviousness of the surface of the analyzed catchment area in the period 2009–2019. On the basis of the curves, it can be concluded that in the period 2009–2019 there was a significant increase in the catchment imperviousness. In the period 2009–2016, the increase in the degree of the imperviousness was about 8% (from Imp = 0.33 to Imp = 0.41), while in the period 2016–2019 the process of growth of impervious surfaces accelerated rapidly. In this period, the share of impervious areas increased from 41% to 55%, which was related to numerous investments in the area under consideration.

The total length of the sewer network is 5583 m, of which the main canal is 1569 m long and its diameter from top to outlet varies between 600–1250 mm.. The main channel receives rainwater from 17 side channels, whose diameters vary in the range of 300–1000 mm. The total volume of pipes with stormwater wells is 2032 m$^3$. The slope of the collector in individual sections varies from 0.04% to 3.90%, while the slope of side channels is 2.61% maximum. The catchment area leveling is 12.0 m and the average land gradient is 7.1%. Stormwater from the catchment area is discharged to the Silnica River, but prior to this is directed to a stormwater treatment plant (STP). The total stream of stormwater through the S1 channel flows into the treatment plant when the filling in the diversion chamber (DC) does not exceed $h = 0.42$ m.

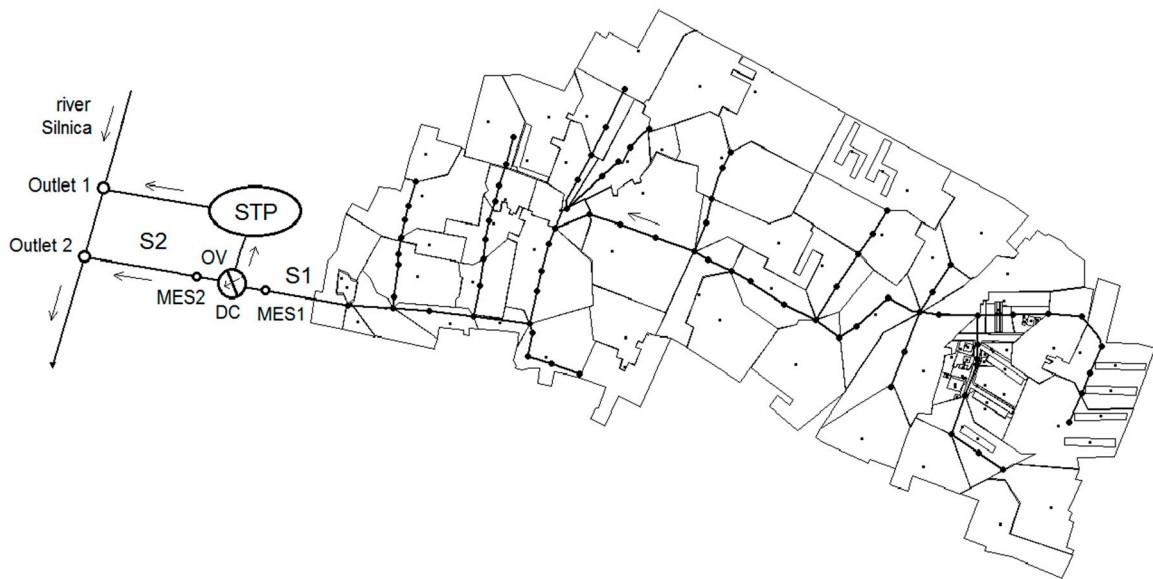

**Figure 2.** Scheme of considered catchment area with main and side channels.

When the filling in the diversion chamber exceeds the permissible value h, the stormwater is discharged via storm overflow (OV) to the S2 channel, through which the stormwater flows into the Silnica river.

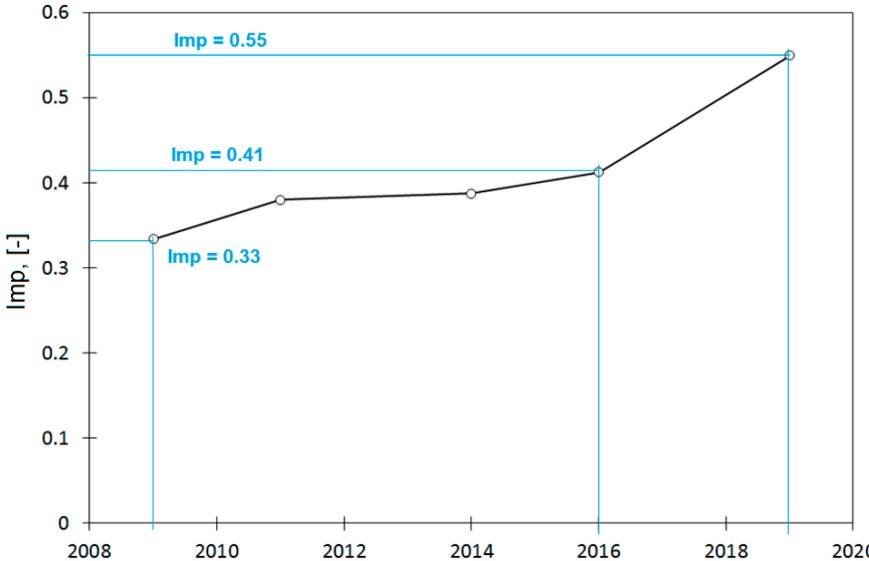

**Figure 3.** Change in the degree of imperviousness (Imp) of the tested municipal catchment area in the period 2009–2019.

In the analyzed catchment area, in the period 2009–2011, continuous measurements of the amount of stormwater flowing out of the catchment area were carried out using the MES1 flow meter located in channel S1 at a distance of 3.0 m from the inlet to the DC chamber. However, since 2015, in the inlet (S1) and discharge (S2) channels flow meters MES1 and MES2 have been installed to measure the values of channels filling and flows. Detailed description of the installed measuring equipment can be found in the paper by Szeląg et al. [50].

## 4. Results

Following the steps of the calculation algorithm (Figure 1), a logit model, a rainfall generator, and a calculation of the annual number of storm overflow discharges for the stormwater network under study were determined for the construction of the probabilistic model (one of the calculation variants considered aspects related to the sustainable way of the development of the catchment area). The results of the calculations obtained are discussed below.

### 4.1. Logistic Regression

On the basis of the measurements of sewer filling, flows, and rainfall in the examined catchment area and stormwater system, rain episodes were separated and their characteristics were determined ($P_{tot}$, $t_r$), and independent variables accounted for in the logit model were determined. On this basis, the empirical coefficient $\alpha k$ in the logit model and the values of the measures of matching the calculation results to the measurements whose numerical values are listed in Table 1 were then determined. On the basis of the data in Table 1, it can be concluded that the determined model of logistic regression is characterized by satisfactory predictive capabilities.

**Table 1.** Summary of the values of $\alpha_k$ coefficients and measures of adjustment of calculation results to measurements (SENS, SPEC, $R_z{}^2$).

| Independent Variables | $\alpha_k$ | Standard Deviation |
|---|---|---|
| Independent word | −6.394 | 0.887 |
| $P_{tot}$ | 0.742 | 0.092 |
| $t_t$ | −0.08 | 0.01 |
| Imp | 8.322 | 1.373 |
| SENS = 88.99% | | |
| SPEC = 82.50% | | |
| $R_z{}^2$ = 85.97 | | |

Note: $P_{tot}$ = rainfall depth; $t_r$ = duration; Imp = degree of imperviousness; SENS = sensitivity; SPEC = specificity; $R_z{}^2$ = calculation error.

In order to complete the calculation results, a graphical interpretation of the SENS = f(SPEC) relation is presented in Figure 4.

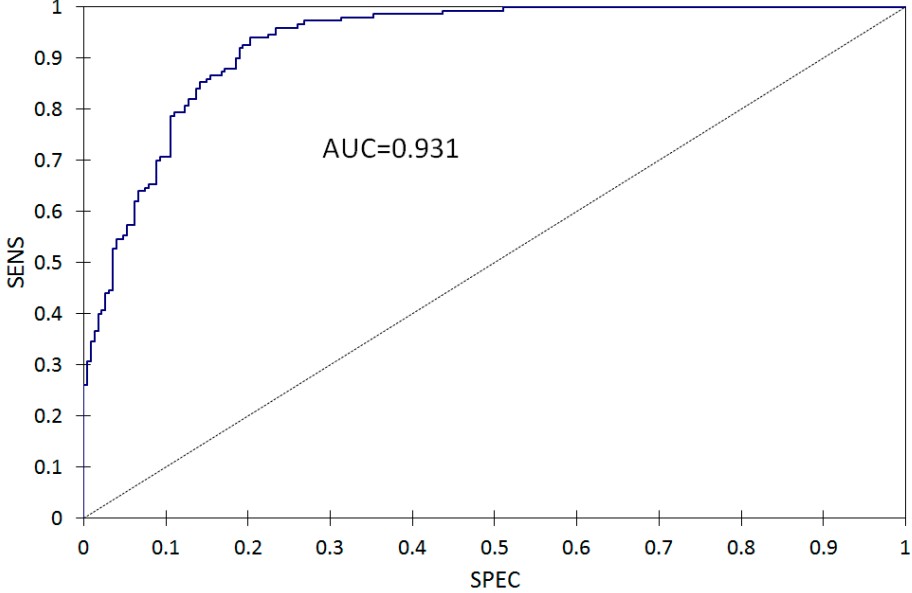

**Figure 4.** Relationship SENS = f(SPEC) obtained o the basis of calculation.

Based on this relationship, the predictive capabilities of the model can be determined by analysing the surface area (AUC) above the SENS = SPEC curve. The closer the SENS value is to one, the more accurate the model is. From the variation of the curve obtained in Figure 4 it can be concluded that the determined model is characterised by a high fit of the calculation results to the measurements, which is confirmed by the obtained value of AUC = 0.931. On the basis of the results obtained, it can be concluded that out of 209 storm overflow discharges observed, the model correctly identified 188 overflow episodes, which is indicated by SENS = 88.99%. On the other hand, out of 240 recorded rainfall events, when no discharge occurred, in 198 cases the results of calculations were similar to the measurements—SPEC = 82.50%. To sum up, on 449 events in 386 episodes the results of the simulation were consistent with the measurements. On the basis of data from Table 1 it is also shown that among the independent variables taken into account in the model, rainfall characteristics ($P_{tot}$, $t_r$) and the degree of imperviousness of the catchment have a significant impact on the occurrence of storm overflow discharge. These results confirm the results of computational experiments [23–25] and studies carried out in urban catchments [51]. The above conclusion is confirmed by Figure 5, which shows exemplary logit curves prepared for $t_r$ = 45 min.

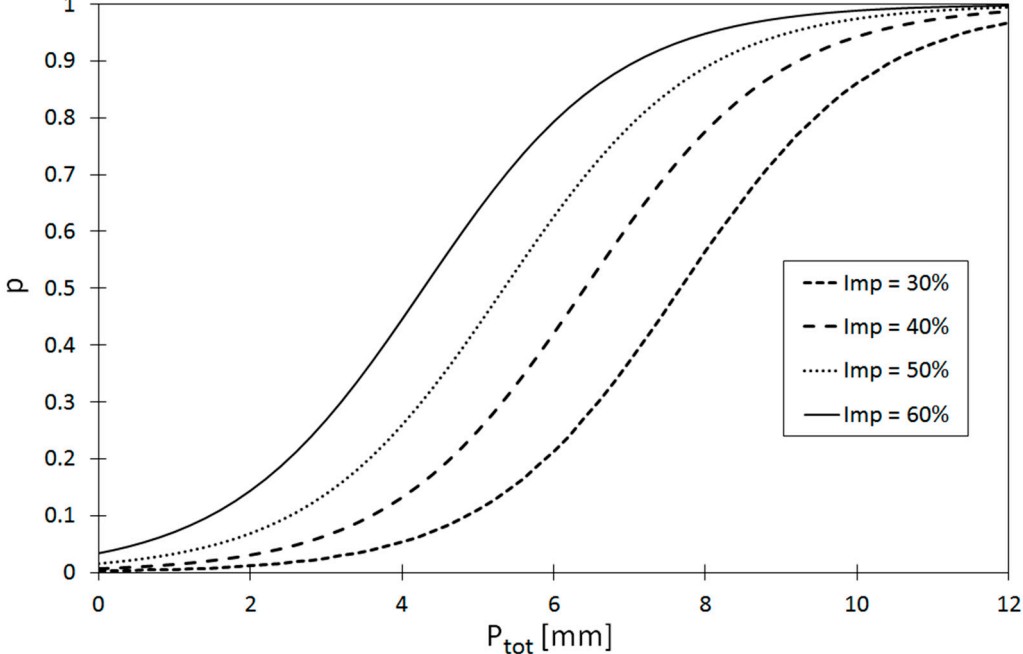

**Figure 5.** Change in the degree of imperviousness of the tested municipal catchment area in the period 2009–2019 (where $p$ is the probability ($p$) of overflow discharge in a single rainfall episode).

Based on the data in Table 1, the following equation can be given to describe the multidimensional space of independent variables describing the occurrence of a storm overflow discharge:

$$0.742 \cdot P_{tot} - 0.080 \cdot t_r + 8.322 \cdot Imp - 6.394 \geq 0 \tag{5}$$

Based on theoretical considerations concerning models for the forecast of storm overflow discharge volume, taking into account the variables $P_{tot}$ and $t_r$ [23,24], successive transformations of Equation (5) were made:

$$P_{tot} - 0.108 \cdot t_r + 11.526 \cdot Imp - 8.617 \geq 0 \tag{6}$$

Assuming the average value of the degree of imperviousness of the catchment in the period 2008–2016 equal to Imp ≈ 40%, one obtains the following expression:

$$P_{tot} - 0.108 \cdot t_r + 4.610 - 8.617 \geq 0 \tag{7}$$

which, after appropriate transformations, can be saved in the form:

$$P_{tot} - 0.108{\cdot}t_r - 4.007 = P_{tot} - 0.108{\cdot}t_r - d_{min} \geq 0, d_{min} \approx 4.007 \tag{8}$$

On the basis of the obtained relation in Equation (8) it can be stated that the calculated value of $d_{min}$ = 4.007 mm conditioning the occurrence of storm overflow discharge differs more than 5% from the value of the weighted average retention of the investigated catchment equal to $d_{av}$ = 3.86 mm.

Thus, the empirical coefficients determined in the logit model have a physical interpretation and the obtained model can be applied in other urban catchments with different physical-geographical characteristics. However, in order to confirm this, further analyses are necessary to confirm the relationships established in the paper. The results obtained in terms of making decisions on the direction of catchment area management and the design of stormwater management systems have a practical significance in terms of reducing the negative impact of the stormwater system on the receiver.

In order to supplement the above considerations and assess the impact of the increase in the degree of land imperviousness on the effect of the storm overflow in the urban catchment area in question, the curves determining the probability of a single discharge for the value of Imp = 25–50% were determined (Figure 6).

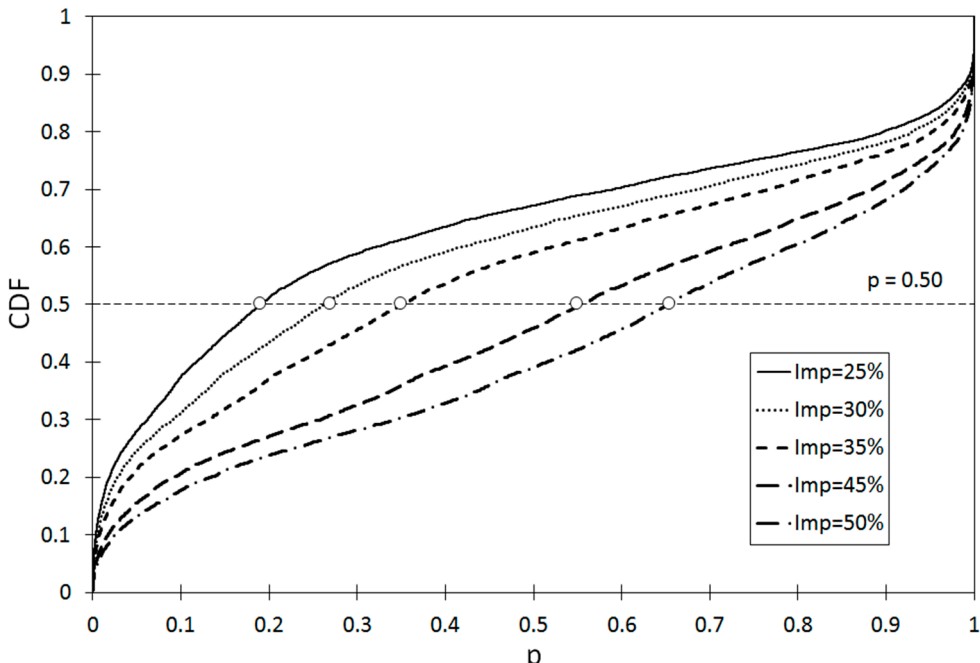

**Figure 6.** Impact of the degree of land imperviousness (Imp) on the probability (*p*) of overflow discharge in a single rainfall episode (percentile values) in the examined urban catchment area (CDF is the cumulative probability density distribution).

The curves determined in Figure 6 confirm the significant influence of the degree of the catchment imperviousness on the occurrence of storm overflow discharge. On this basis it can be concluded that with the increase in urbanization of the area, the values of probability of discharge, mainly in relation to the value of the percentile *p* = 0.50, show a strongly increasing tendency. For Imp = 50%, the average probability of a discharge by storm overflow in a single rainfall episode is 0.63.

## 4.2. Annual Number of Overflow Dicharges and Reduction Their Number

In order to verify the predictive capabilities of the logit model obtained, the annual number of discharges in individual years (2008–2017) was calculated and compared with the measurement results (Table 2). Based on the data in Table 2 it can be stated that the results of the annual simulation

of the number of discharges obtained with the model and determined from the measurements are characterized by a high convergence. This confirms the difference in the number of discharges between the values obtained from the measurements and the calculated values, which amounts to 3 and does not exceed 15%.

**Table 2.** Comparison of the annual number of discharges obtained from measurements ($Z_{mes,a}$) and calculated with the logit model ($Z_{logit,a}$).

| Year | Imp$_{act}$ | N | $Z_{mes,a}$ | $Z_{logit,a}$ | % Imp (Control) | |
| --- | --- | --- | --- | --- | --- | --- |
| | | | | | Imp, % | $Z_{c,logit,a}$ |
| 2008 | 33 | 43 | 15 | 13 | 33 | 13 |
| 2009 | 33 | 47 | 16 | 15 | 33 | 15 |
| 2010 | 35 | 47 | 18 | 16 | 33 | 13 |
| 2011 | 38 | 51 | 20 | 18 | 33 | 12 |
| 2012 | 38.3 | 36 | | 17 | 33 | 12 |
| 2013 | 38.6 | 41 | | 18 | 33 | 14 |
| 2014 | 39 | 44 | | 24 | 33 | 18 |
| 2015 | 40 | 58 | 24 | 21 | 33 | 13 |
| 2016 | 41.3 | 44 | 20 | 21 | 33 | 15 |
| 2017 | 45 | 38 | 20 | 19 | 35 | 12 |
| 2018 | 50 | 42 | 20 | 21 | 40 | 16 |

Note: $N$ = number of rainfall events per year; $Z_{mes}$ = the measured number of overflow discharges per year; $Z_{logit}$ = the calculated (logit model) number of overflow discharges per year; $Z_{c,logit}$ = the calculated number of overflow discharges per year (control, land use-management).

Taking into account the results of calculations obtained and the usefulness of the determined logit model for the simulation of the annual number of discharges, the reduction of the degree of imperviousness of the catchment area in the years 2009–2016 was considered to be 33%, in 2017 to be 35%, and in 2018 to be 40%. The paper assumes that the reduction of the degree of the catchment imperviousness in the successive years is based on the introduction of solutions allowing the reduction of the stream of outflowing stormwater by designing green roofs [6,16] on buildings (residential, public utility, etc.), replacing impermeable surfaces of parking lots with other types of surfaces [52,53], intercepting rainwater within individual objects in barrels [16,18], and draining rainwater arising within the property to drainage systems [18,27,54]. The above mentioned solutions lead to an increase in the retention of the catchment area, which is confirmed by computer simulations and field studies conducted on a long-term basis. However, due to the complex relationships between the characteristics of the adopted solutions, which affect the variability of rainwater outflow in time, some simplifications are applied in simulation calculations [14]. Although the results obtained in this way differ to a certain extent from the actual variability of the imperviousness stream, from the point of view of engineering considerations they constitute a compromise between the accuracy of the calculation results and the complexity of the model created.

### 4.3. Probabilistic Model

Calculations of the number of storm overflow discharges (Table 2) carried out for the catchment area under consideration in individual years (2008–2018) have shown that the reduction of the degree of the catchment imperviousness, and thus the balanced development of the catchment area, has a significant impact on the reduction of the annual number of overflow discharges, which has an impact on the improvement of the quality of the receiver waters and the state of ecological equilibrium. This fact is confirmed by the works of numerous researchers conducting field studies and performing computer simulations [1,55–57]. In the catchment in question, the reduction of the degree of land imperviousness in the successive years by 2–10% leads to a decrease in the annual number of overflow discharges from $Z_a = 2$ to 8 (Figure 7). Due to the random nature of rainfall, the above simulations

were supplemented by the determination of probabilistic solutions expressing the number of storm overflow discharges for subsequent degrees of land imperviousness (Figure 7).

The curves determined in Figure 7 confirm the significant influence of the land imperviousness degree on the annual number of storm overflow discharges. For example, the reduction of the imperviousness degree from Imp = 40% to Imp = 35% leads to a decrease in the average number of overflow discharges ($p = 0.50$) from $Z_a = 22$ to 19. At the same time, analysing the results obtained above, it can be noted that the ranges of variation of the $Z$ value in Table 2 are within the range of probabilistic solutions, which confirms the usefulness of the model proposed in the paper to simulate the annual number of overflow discharges. In order to supplement the conducted analyses, a simulation of the impact of the dynamics of changes in the degree of imperviousness of the catchment area on the annual number of discharges was carried out. For this purpose, the urbanization of the catchment was analyzed for the period of two years, assuming the initial degree of imperviousness as equal to 33%. In the analyses two variants of the catchment urbanization were considered. In the first variant, the changes in the degree of the catchment imperviousness in the following years were assumed to be equal to 1% and 7%, respectively, whereas in the second variant, the changes amounted to 4% and 4%, respectively. The results of the calculations are shown in Figure 7.

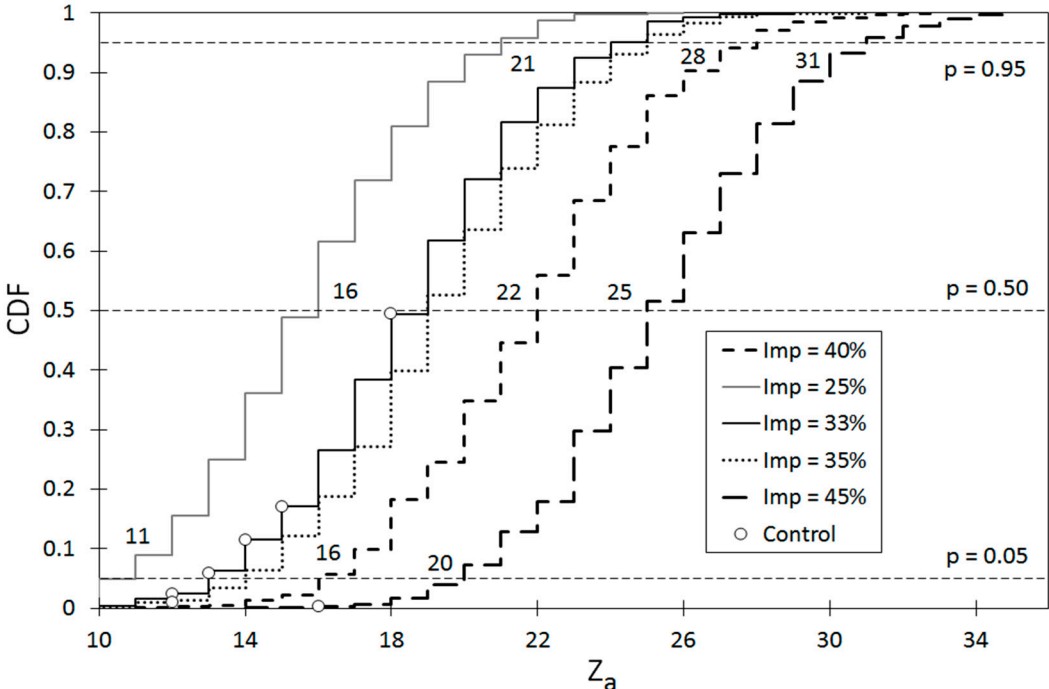

**Figure 7.** Empirical cumulative probability density distribution (CDF) describing the probability of exceeding the annual number of discharges ($Z_a = Z_{c,logit,a}$) for different degrees of catchment imperviousness (Imp).

In order to supplement the conducted analyses, a simulation of the impact of the dynamics of changes in the degree of imperviousness of the catchment area on the annual number of discharges was carried out. For this purpose, the urbanization of the catchment was analyzed for the period of two years, assuming the initial degree of imperviousness as equal to 33%. In the analyses two variants of the catchment urbanization were considered. In the first variant, the changes in the degree of the catchment imperviousness in the following years were assumed to be equal to 1% and 7%, respectively, whereas in the second variant, the changes amounted to 4% and 4%, respectively. The results of the calculations are shown in Figure 8. On the basis of the determined curves (Figure 8) it can be concluded that despite the identical target degree of the catchment imperviousness in both variants (up to 41%), in the second variant a slightly higher number of storm overflow discharges was obtained.

The above calculation example shows that the dynamics of the urbanization process have a significant impact on the number of overflow discharges in a long-term period (in this case two years). From the point of view of the use of the model proposed, this is an important piece of information that may, in practice, have an impact on the way in which the catchment area is developed, and consequently, on the operation of the stormwater system and the condition of the receiver.

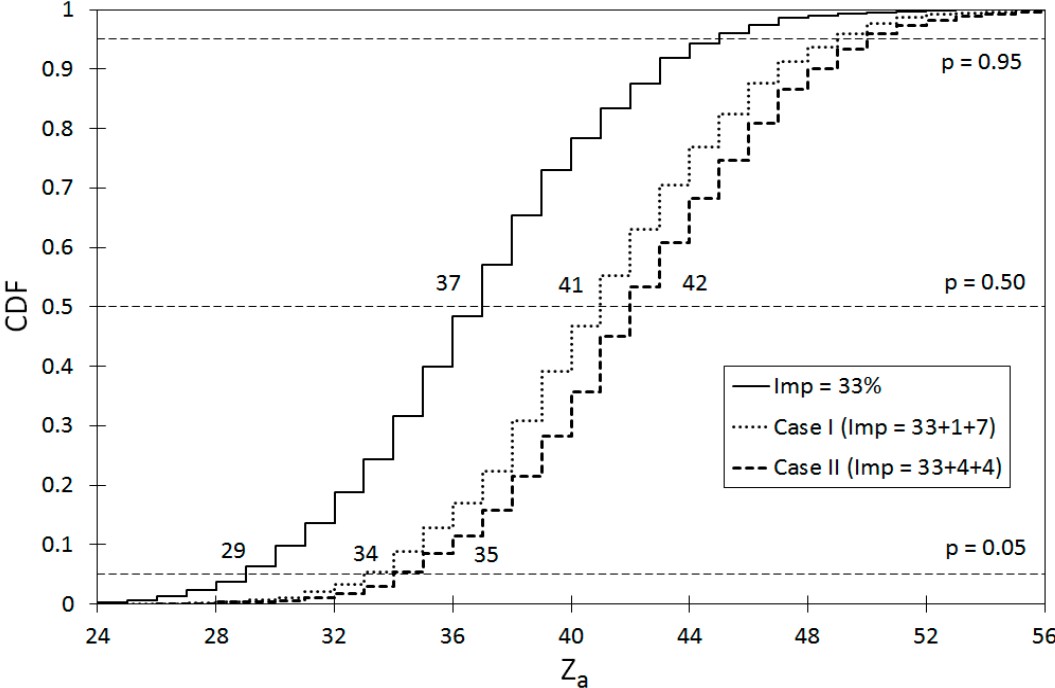

**Figure 8.** The influence of dynamic changes in the degree of the catchment imperviousness on the number of storm overflow discharges over the two-year period ($Z_a = Z_{c,logit,a}$). Note: Imp = degree of imperviousness; $Z_a$ = annual number of discharges.

The results of the calculations presented above are important, because on their basis (at the stage of selecting the variants of the catchment area expansion) it is possible to select the variant for which the total number of discharges will be the smallest, which allows limitation of the impact of the stormwater system on the receiver waters.

## 5. Conclusions

The calculations performed showed that the proposed probabilistic model is a valuable tool for simulating the annual and long-term number of storm overflow discharges with the assumption of the stochastic character of precipitation and the imperviousness degree of the catchment area. The proposed approach, in comparison with other authors, is a significant development of the subject, because the developed model allows, on the one hand, forecasting of the number of discharges, and on the other hand, reduction of the number of discharges through proposals for appropriate land use.

The paper shows that using the logit model it is possible to simulate the operation of a storm overflow, having only the total depth of rainfall, its duration, the degree of imperviousness of the catchment area, and the catchment retention. The obtained results are of great practical importance, as they enable adaptation of the logit model to other catchments.

The calculations carried out and discussed in the paper also showed that the proposed probabilistic model allows one to forecast the influence of the dynamics of changes in the degree of the catchment imperviousness on the number of discharges in the assumed period of time in a long-term perspective. This is an important advantage of the model, because in practical considerations it can be treated as a tool for planning sustainable development of catchment areas (by designing rainwater management

systems and appropriate selection of surfaces for roads, parking lots, etc.) in such a way as to limit the negative impact of the catchment urbanization on the receiver waters.

**Author Contributions:** Conceptualization, B.S.; methodology, B.S., A.C., and S.S.; software, B.S., A.C., and S.S.; formal analysis, B.S., A.C., and S.S.; investigation, B.S., A.C., S.S., and J.T.G.; writing—original draft preparation, B.S., A.C., S.S., J.S., and J.T.G.; writing—review and editing, B.S., A.C., S.S., J.S., and J.T.G.

**Funding:** The work was founded by the Polish Ministry of Science and Higher Education, the RID (Regional Excellence Initiative) project, according to the agreement: 025/RID/2018/19 of 28/12/2018 with total budget of 12,000,000 PLN.

**Conflicts of Interest:** The authors declare no conflict of interest.

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
