# Peer review of "Urbanization and Management of the Catchment Retention in the Aspect of Operation of Storm Overflow: A Probabilistic Approach"

_sustainability, doi:10.3390/su11133651_

Round 1

Reviewer 1 Report

This is an interesting and in generally clearly written manuscript. I would suggest only few changes to make it more reader friendly.

Figure 1 shows the calculation algorithm and the calculation steps are described below. I would suggest to mark “a” to “f” sectors on the fig. 1

Lines 246-247 – the sentence is not full, please check.

Line 270 – please add the name of the country

Lines 271-272 – could You please add the information about the share of different areas and where are they located?

There are 2 figures 5 (page 11 and page 13)

Table 1. even if symbols are explained before, eg. line 187, please add this explanation below the table. The same for Fig 4, 5 etc. In my opinion the description of the figures should follow the description used in case of fig 5 (page 13)

Author Response

Figure 1 shows the calculation algorithm and the calculation steps are described below. I would suggest to mark “a” to “f” sectors on the fig. 1

Answer: Fig. 1 shows the markings used in the manuscript.

Lines 246-247 – the sentence is not full, please check.

Answer: The sentence has been corrected

Line 270 – please add the name of the country

Answer: The text contains information about the location of the analysed catchment area.

Lines 271-272 – could You please add the information about the share of different areas and where are they located?

Answer: The text contains information on the share of particular areas in the catchment area.

There are 2 figures 5 (page 11 and page 13)

Table 1. even if symbols are explained before, eg. line 187, please add this explanation below the table. The same for Fig 4, 5 etc. In my opinion the description of the figures should follow the description used in case of fig 5 (page 13)

Answer: Below the tables and most of the drawings there are descriptions of the following symbols

Reviewer 2 Report

Line 202 - To build a logistic regression model the period 2009-2011 is too short. Why this period?

Line 215 –What are the criteria for selecting the period (2008-2017)

Line 274 - refers to the analysis period 2008-2019, but in Line 275 it refers 2009-2019

Improve catchment characterization

In Figure 3, the Y- axis unit is missing. What is referred to in the text on this figure is not perceived. What is observed in the figure does not correspond to the values indicated in the text (line 273-278). Figure 3 is too large.

Figure 5 –  indicate the meaning of CDF (Y-axis)

There are two figures with the number 5. Review the numbering of the figures and the references in the text.

Figure 6 –  indicate the meaning of CDF (Y-axis) and the meaning of Za (X-axis)

Cases I and II indicated in figure 6 should be referred in the text.

Author Response

Line 202 - To build a logistic regression model the period 2009-2011 is too short. Why this period?

Answer: In the years 2009 - 2011 continuous flow measurements were carried out. Data from the period 2012-2017, i.e. 261 episodes, were used to construct the model.

Line 215 – What are the criteria for selecting the period (2008-2017)

Answer: In the period 2008 - 2017 the measurements of rainfall, flows and canal filling were carried out. From 2008 to the present day, rainfall and flow measurements have been carried out with the use of flow meters MES1 and MES2.

Line 274 - refers to the analysis period 2008-2019, but in Line 275 it refers 2009-2019

Answer: The remark has been included in the manuscript.

Improve catchment characterization

Answer: The description of the catchment area has been extended.

In Figure 3, the Y- axis unit is missing. What is referred to in the text on this figure is not perceived. What is observed in the figure does not correspond to the values indicated in the text (line 273-278). Figure 3 is too large.

Answer: The units have been completed in the figure. The figure shows the numerical values that are included in the text.

Figure 5 – indicate the meaning of CDF (Y-axis)

Answer: The title of the figure 5 has been corrected.

There are two figures with the number 5. Review the numbering of the figures and the references in the text. Figure 6 – indicate the meaning of CDF (Y-axis) and the meaning of Za (X-axis)

Cases I and II indicated in figure 6 should be referred in the text.

Answer: The numbering of the figures has been adjusted.

Reviewer 3 Report

The authors present an integrated framework to evaluate the impact of different imperviousness in a catchment area on probability of storm overflow in a purely statistical way which is easy to apply because of avoiding physically-based flood models. This might be useful for decision makers in the management of urban catchment areas. Overall ideas are clear. However, to make readers understand this method is reliable, in my opinion, you need to add more evidences which ensure the reliability of the results obtained in this study, because all the elements are composed not of physical models but of statistical models. Please address the following comments to make this work satisfactory for its publication.

L 305

I understand this section purposes to obtain a relation between Ptot and tr (Eq.8). In spite of replacing imperviousness with a constant value in the end, why do you need to include that in regression analysis?

Table 1

Please add figures showing relation between each independent variable and the estimated probability of storm overflow v.s. the observed probability. Only indexes SENS, SPEC, and R2 are not sufficient to show the validity of the applied model.

Some minor comments are as follows.

Title

You should add “of” between “management” and “the catchment retention”

Overall

The word “impervious”, which is actually an adjective, is used as a noun in many parts of the manuscript. Please replace it with “imperviousness”

Eq. (1)

A summation in the exponent in the denominator is not up to j + j but up to i + j?

L 179

Index i is confusing. In Eq. (2), index “i” shows a particular rainfall event (in L 179), but “i” on the next line (180) in A1, A2, …, Ai shows the number of independent variables for catchment characteristics. Use different index for a particular rainfall event in Eq. (2).

L 185

Equation (3) should not be an equation but an inequality (and the left hand side should be larger than the right one, to assure p in Eq. (1) will not be lower than 0.50)?

L 340

Please add explanation of how you can assume that Ptot – 0.108*tr (= 4.007) corresponds to the retention in the catchment area.

L 364

“the average probability of a discharge” à “the average probability of a discharge by storm overflow”?

L 453

Remove “in discussed”

Author Response

L 305: I understand this section purposes to obtain a relation between Ptot and tr (Eq.8). In spite of replacing imperviousness with a constant value in the end, why do you need to include that in regression analysis?

Answer: The recorded relationship did not reflect the relationship received and was corrected in the manuscript.

Table 1 Please add figures showing relation between each independent variable and the estimated probability of storm overflow v.s. the observed probability. Only indexes SENS, SPEC, and R2 are not sufficient to show the validity of the applied model.

Answer: In order to visualize the obtained results of calculations in the paper, the SENS = f(SPEC) curve has been placed on the basis of which predictive capabilities of the obtained model can be evaluated.

You should add “of” between “management” and “the catchment retention”

Answer: The title of the article has been corrected.

The word “impervious”, which is actually an adjective, is used as a noun in many parts of the manuscript. Please replace it with “imperviousness”

Answer: The note is included in the manuscript.

Eq. (1) A summation in the exponent in the denominator is not up to j + j but up to i + j?

Answer: The note is included in the manuscript.

L 179 Index i is confusing. In Eq. (2), index “i” shows a particular rainfall event (in L 179), but “i” on the next line (180) in A1, A2, …, Ai shows the number of independent variables for catchment characteristics. Use different index for a particular rainfall event in Eq. (2).

Answer: The symbols have been corrected in equations

L 185 Equation (3) should not be an equation but an inequality (and the left hand side should be larger than the right one, to assure p in Eq. (1) will not be lower than 0.50)?

Answer: The dependence given in the manuscript has been corrected

L 340 Please add explanation of how you can assume that Ptot – 0.108*tr (= 4.007) corresponds to the retention in the catchment area.

Answer: The dependence given in the manuscript has been corrected

L 364 “the average probability of a discharge” à “the average probability of a discharge by storm overflow”?

Answer: The remark has been included in the manuscript.

L 453 Remove “in discussed”

Answer: The remark has been included in the manuscript.

Round 2

Reviewer 2 Report

No comments